# High-Index Glass Ball Retroreflectors for Measuring Lateral Positions

**DOI:** 10.3390/s19051082

**Published:** 2019-03-03

**Authors:** Andrea Egidi, Alessandro Balsamo, Marco Pisani

**Affiliations:** INRiM—National Institute of Research in Metrology, Division of Applied Metrology and Engineering, 10135 Torino, Italy; a.balsamo@inrim.it (A.B.); m.pisani@inrim.it (M.P.)

**Keywords:** backscattering, retroreflectors, glory, ray tracing, high index ball lenses

## Abstract

This paper is concerned with backscattered luminous signals, coming from a particular class of dielectric spheres illuminated by a coherent source. The purpose is to measure the lateral position of the sphere serving as an optical target, to achieve an overall contactless sensor of lateral position in space. Traditional approaches and theories such as ray-tracing and Mie scattering—as implemented in dedicated software—are applied to investigate their fitness for purpose in this application. No previous literature was found dealing with this specific case. Unfortunately, our observations did not match the theories’ predictions to an acceptable degree, and these approaches proved to be unsatisfactory. The rest of the paper focusses then on the development and comparison of suitable algorithms to compute the image coordinates of a representative point, which was in fact the true motivation of this work. Two original algorithms are proposed and discussed. Their robustness and repeatability are benchmarked under noisy conditions and at different distances from the target, with simulated as well as real images. Both resulted capable of sub-pixel accuracy.

## 1. Introduction

High-index ball lenses are widely used in optics-oriented applications, e.g., in coupling fibers and—more generally—in focusing or collimating light beams. This paper is centred on their peculiar property of retroreflecting the light. Several alternative retroreflector types ara available based on different principles, such as glass or hollow corner cubes [1,2], cat’s eyes, and Spherically Mounted Retroreflectors (SMRs). One of the main disadvantages of these retroreflectors is a rather limited acceptance angle, restricing the illuminating beam to impinge from a narrow solid angle. To overcome this, glass spheres were introduced in the 1960s [3]. More recently, glass spheres with refracrive index *n* = 2 and thin supporting stems were introduced [4], to enlarge the acceptance angle to a nearly full solid angle (apart from the stem) and to exploit their (nominal) property of ideal retroreflection of light (at least under certain circumstances, see the next section for more details). However, they suffer poor efficiency as most of the impinging light is refracted through and is wasted on the rear: as little as approximately 15% of the light power is retroreflected in fact. Such spheres are available on the market up to a limited size (typically 8 to 10 mm diameter) as they are usually machined out of a single glass sheet. The introduction of these spheres triggered many studies available in literature, among which a detailed ray-tracing interpretation of their retroreflection [5].

Retroreflectors can be used for two different purposes: to measure either the *longitudinal distance* or the *later position* of a target to a reference point. Interferometric measurements are a typical example of the former use, while laser trackers [6,7] combine the two, effectively measuring the 3D spherical coordinates of a target point. The mentioned studies focussed on the longitudinal distance [4], while none investigated in full the ultimate limit of the lateral position detection. In addition, the actual distribution of the retroreflected light—detected as an image by a camera—has not been investigated yet.

The work described in this paper was intended to improve the performance of a novel coordinate measurement concept referred to as InPlanT [8]. In it, a *n* = 2 glass sphere is used as retroreflecting target, whose image is acquired by a camera. The sphere lateral position is detected and measured by the 2D (local) coordinates of the image in the camera system of coordinates. The algorithmical reduction of the image to a representative point is key and any improvement in the algorithm would impact straight to the overall InPlanT measurement accuracy. An apparently promising approach was by fitting the actual image against a template image as predicted by a model equation, which was not available though. This motivated a theoretical/experimental study to benchmark available theories in literature, from classical ray-tracing to the exotic theory of the *glory*. Unfortunately, none proved adequate for this specific case. As the original intent was to improve a localisation algorithm, we eventually forewent the image template approach and focussed on the only available first-principles characteristic, namely the intrinsic image central symmetry naturally stemming from a sphere.

In the end, this paper describes a method for measuring the lateral position of a *n* = 2 sphere target. Such measurement originated from a specific project (where the *x*-*y* coordinates of the center of the sphere needed to be carefully estimated to close the control loop of a tracking system), but it is of wider applicability, i.e., in non contact measurements in a view plane. This is a common problem to all camera based sensors. The knowledge of the actual beam pattern retroreflected by the target may be beneficial also when the light detection is done by means other than cameras, as it helps designing the optical set up.

The next section decribes the optical set up used in our investigation. Section 3 reviews exisiting theories in the attempt of predicting the actual image. Section 4 describes the developed algorithms. Some conclusions follow in Section 5.

## 2. Experimental Set Up

We used 16 mm diameter spheres made of Ohara S-LAH79 *n* = 2 glass (Ohara Corporation, Rancho Santa Margarita, CA, USA), illuminated by collimated coherent light; the spheres were kindly provided by the National Physical Laboratory (NPL, Teddington, Middlesex, UK); the complete optical characteristics of their glass can be found in the Ohara reference page, while the refractive index was verified on the site refractiveindex.info). The setup used to study the physics of the *n* = 2 spheres is shown in Figure 1. A 20 mW fiber coupled red laser crosses a polarizer and is collimated by means of a converging lens, then directed towards the sphere; the backscattered beam is then deviated at 90° through a non-polarizing cubic beam splitter, and impinges on the active area of a CMOS camera; the acquired images, afterward, are post-processed to extract the information needed. In Figure 2 a typical pattern of the backscattered signal is shown: it is generated by a 8 mm radius sphere located at a distance of 254 mm far from the camera used to collect the images; our main purpose was to define a viable and repeatable method to gather information on the lateral positions of the sphere itself.

Also keeping the kind of glass fixed, the ring-shaped pattern revealed itself to be strongly dependent on some geometrical and physical factors, accounting for the sphere diameter, the reciprocal distance between sphere and camera, the accuracy of the collimation of the impinging beam, the portion of the sphere that is illuminated, the shape of the wavefront etc. More specifically, the thickness and the angular spacing of the rings is subject to dramatic variation when the sphere is moved along the direction joining the source of the signal to the center of the sphere; correlating such morphological parameters to the position of the sphere in space is the key factor for developing a metrological approach when using this kind of retroreflector in applications where the lateral displacement of a target is the measurand.

## 3. Review and Investigation of Existing Models

Some different strategies have been adopted to interpret the signals and infer the needed information: a classical *ray-tracing approach*, the more general *Mie scattering theory* and the models developed upon the Van de Hulst dissertation on light scattering by particles; anyway, all of them turned out to be unsatisfactory in explaining what happens in our particular situation, so a phenomenological and quantitative study of the peculiar pattern of the signals through *two newly developed algorithms* has been carried out, with the aim of extracting the localization of the target point.

### 3.1. Ray-Tracing

Most of the calculations are based on a few but elegant geometric optics concepts easily found in literature [5]. In the following Figure 3, Figure 4, Figure 5 and Figure 6, the main optical properties of the spheres can be inferred, basing the calculations on the same principles. In Table 1, a list of the fundamental equations accounting for a geometrical interpretation of the backscattered signal is provided (same nomenclature as shown in Figure 3):

#### Ray-Optics on the Sphere at our Disposal

In our specific situation (glass used: Ohara S-LAH79, with *n* = 1.9957745 @ 635 nm, *T* = 20 °C and *P* = 1 Atm), numerical analyses with realistic refraction data have been carried on. Adopting the aforementioned nomenclature, the following relations can be derived:
*Maximum deflection*: χ*_m_* = 0.212 mrad when *ϕ_m_* = 75.091 mrad, that is when *b = b_m_* = 0.599 mm;*Null deflection*: χ = 0 when *ϕ* = *ϕ_0_* = 130.061 mrad, that is when *b = b_0_ =* 1.037 mm (situation of “perfect” retro-reflection).

From Figure 5b it can be argued that, according to the ray-optics, three distinct regions in the codomain of the mapping function *r(b, z)* exist: the first (“3-rays region”), mapped by the impact parameter range [*–b_m_, b_m_*], in correspondence of which, at a certain ray mapped on the screen, three different rays (with three distinct impact parameters) can impinge on the sphere giving the same *r*(*χ*, *z*); the other two regions, symmetrical to the central field, where there is a 1:1 correspondence between a ray impacting on the sphere and a scattered ray mapped on the screen. In Figure 6 a 3D representation of the same concept (changing χ with *b*) can be seen.

Table 2 shows the geometrical properties of the ray-tracing simulation, in input and in output. Figure 7b justifies the choice of S-LAH79 glass for our purposes: the plot clearly shows that the deflection of a beam impacting on a sphere made with this glass changes quite slowly with respect to the impact parameter, inside a reasonable range (*low ray deflection zone*: |χ| < 10 mrad for *b* < 0.35·*R*), without requiring an unpractically low *b*/*R* ratio, whereas the other glasses suffer from a greatest angular spread of the exiting beam, at the same radius of the sphere.

### 3.2. Mie Scattering: A Simple Software Approach

On the web, many software programs aimed to solving the problem of scattering of electromagnetic signals by spheres/spheroids can be found [10,11] and tested; among them, we decided to give preference to the package PyMieScatt [12], implemented in Python, and to the software MiePlot [13]. The angular functions of interest were, respectively, the *parallel*, *perpendicular*, and *unpolarized* scattered intensities *SR(θ)*, *SL(θ)* and *SU(θ)*, with a focus on the backscattering region (*θ*~180°), that is the region where the phenomenon of the *glory* makes its spectacular appearance (Figure 8 and Figure 9). The literature about glories is quite rich [14,15,16], but the most recent outcomes on the topic derive from postulating the existence of surface waves travelling around the outside of the sphere [17,18]. In our case, we deal with a sphere with a refractive index very close to 2, and this means that backscattering is something correlated by geometric rays passing through a huge (with respect to the impinging wavelength) sphere and then suffering at least one internal reflection before travelling back towards the light source; so, simply speaking, the empirical approach does not require us to threat the phenomenon as a glory in a strict sense. In this paper, indeed, we are not interested in elaborating a mathematical model capable of explaining the observable scattering patterns produced by retro-reflecting spheres, but rather to develop a viable artifice to interpret our peculiar class of patterns—and consequently infer metrological information about the coordinates of a target—that, to our knowledge, have not been described before in literature.

## 4. Novel Algorithms for Center Detection

By using both of the computational approaches described in the previous paragraph, we were not able to find a useful correspondence with our intensity patterns: on the one hand, ray-tracing concepts help in predicting the existence of an “accumulation” zone—only one ring where we have many of them—for the rays scattered by the sphere very close to the 180° region (the *primary rainbow*, as stated in Table 2, corresponding to *n_r_* = 1); on the other hand, the Mie approach has shown to be affected by two issues:
(1)The simulated backscattering pattern (Figure 8b) clearly shows a monotonically decreasing envelope (with a hyperbolic trend, according to [17] (ch.13)) of the ripples, starting from the 180° zone, whereas we can infer the presence of a sort of modulating function that breaks the regularity of the envelope (see plot in Figure 15 below).(2)No divergence in the angular spread of the rings, in the acquired images, should exist; this fact is manifestly contradicted analyizing the patterns of the signals at different distances (see plot in Figure 16b below).

For these reasons, we decided to adopt a strictly computational strategy to infer the “centers” of the acquired images based on the only first principle property of the image we could rely on, i.e., the central symmetry of a spherical image. To do so, the cartesian space is convertedto a polar space. The transformed image is then evaluated according to two different criteria implemented in Python, as described in the following sections. Both proved to achieve sub-pixel accuracy. 

### 4.1. Variance-Based Algorithm

This version of software is built on the concept of the minimum value of the sum of the variances per column of the pixel intensities in the polar-converted input images of the backscattered signal (block diagram shown in Figure 10).

The algorithm branches off into two different methods, labeled as “Automatic” or “Forced” procedures. In the former modality, through a subroutine that helps in recognizing the central disc of the acquired image (**1A**), it starts finding the coordinates (y_start, x_start) of the point of maximum brightness (**2A**), that locates the region where the maximum of the scattered energy is stored on the detector; some explicative and simplified lines of code will follow:

img64_float = source.astype(np.float64)
a = np.array(img64_float)
data = np.unravel_index(a.argmax(), a.shape, order=‘C’)
y_start = int(data[0])
x_start = int(data[1])

The next step (**3A**) is determining a squared region on interest (“*ROI*”, whose odd side length is equal to “*span*”), centered on the brightest pixel of the image, that will be the “envelope” of the points defining the centers (subsequently cycled in a double loop) of what we have defined “*sensitive areas*” (**4**), exploring the whole image:
span = int(np.ceil(kp.size/(4*np.sqrt(2)))*2 + 1)

In the automatic procedure, kp.size is the diameter of the blob found by the OpenCV [19] *SimpleBlobDetector* class: in this formulation, span is equal to half of the side of the maximum square inscribed in the central bright disc of the image; in the forced procedure (**1B**), this parameter is simply replaced by an arbitrary quantity, evalutaed in relation with the central disc size (this sometimes reduced the computation time during the testing process). An appropriate cropping of the original image, then, avoids inconsistencies when dealing with situations when the pattern of the signal is very asymmetrical inside the frame:

width_source, height_source = img.size
crop_width = 2 * min(x_start, width_source - x_start, y_start, height_source - y_start)
crop_height = crop_width 
cropped_img_temp = img.crop((x_start - crop_width/2, y_start - crop_height/2, x_start + crop_width/2, y_start + crop_height/2))

sensitive_area_side = crop_width - span


The next step consists in the cartesian to polar conversion of the image (**5**) through the OpenCV *linearPolar* class, cycling the poles of the transformations through all *x* and *y* coordinates of the ROI; the parameter investigated (our “probe”) is the sum of the variances, per columns, of the pixel intensities (**6**):
results = []
range_x = xrange(x_start_new-(span-1)/2,x_start_new+(span+1)/2,1)
range_y = xrange(y_start_new-(span-1)/2,y_start_new +(span+1)/2,1)
for x, y in itertools.product(range_x, range_y):
 Center = (x, y) 
 polar_image = cv2.linearPolar(img64_new_float, Center, sensitive_area_side/2, cv2.WARP_FILL_OUTLIERS)
 polar_image_norm = polar_image / 255
 im = polar_image_norm
 column_variance = np.var(im, axis = 0)
 var_tot = np.sum(column_variance)
 results.append(var_tot)


The iteration order *n* (with 1 < *n* < *span*^2^) where the minimum of the sum of the variances per columns is found, finally, identifies the pole of the polar transformation corresponding to the best candidate as “center” of the original image (**7**). The sub-pixel accuracy is obtained by least squares fitting the 3D matrix of the variances inside the ROI area with a rotated paraboloid (Figure 11a):
matrix_variances = np.transpose(np.resize(results, int(span)**2).reshape(int(span), int(span)))
def parab(X, amp, shift_x, shift_y, cost):
 x, y = X
 return amp * (x**2 + y**2) + shift_x * x + shift_y * y + cost
p0 = 5000, -38000, -45000, 20000 
popt, pcov = curve_fit(parab, (x,y), z, p0)
amp = popt[0]
shift_x = popt[1]
shift_y = popt[2]
cost = popt[3]


### 4.2. FFT-Based Algorithm

This second version of the software is built on the concept of the minimum value of the sum per column of the pixel intensities of the FFT-transformed pictures of the polar-converted original images (block diagram shown in Figure 12).

In this more sophisticated arrangement, the strategy followed by the FFT-based algorithm is the same as before in its first steps (**1**–**5**), but the basic concept, here, is computing the 2D discrete Fourier Transform of a bidimensional array (our polar-transformed image) by means of the Fast Fourier Transform (Numpy function *fft.fft2*) (**6**):
for x, y in itertools.product(x, y):
 Center = (x, y) 
 polar_image = cv2.linearPolar(img64_new_float, Centro, sensitive_area_side/2, cv2.WARP_FILL_OUTLIERS)
 polar_image_norm = polar_image/255
 f = np.fft.fft2(polar_image_norm)
 fshift = np.fft.fftshift(f)
 magnitude_spectrum = (np.abs(fshift))
 height = magnitude_spectrum.shape[0]
 width = magnitude_spectrum.shape[1]
 cropped_FFT_up = magnitude_spectrum[0:height/2 - 1, 0:width] 
 cropped_FFT_down = magnitude_spectrum[height/2 + 1:height, 0:width] 
 int_pix_no_DC = np.sum(np.power(cropped_FFT_up, 2) + (np.power(cropped_FFT_down, 2))) 
 results.append(int_pix_no_DC)


The hearth of this portion of the algorithm is the removal of the central row of the raw power spectrum, that is the DC component of the FFT: choosing a “bad” pole, in the double loop, for the cartesian to polar transformation, in fact, induces a periodic perturbation of the vertical bands of the polar image (adding a spurious component to the spectrum of the frequencies of the image), as much important as the further we go away from the point of maximum symmetry. The choice of the point that minimizes the sums of the squared intensities per column—the probe of this version of the algorithm—of the raw power spectrum (deprived of the central row) of the polar-transformed image defines, therefore, the final coordinates of our “center” (Figure 13). The sub-pixel accuracy is then gained like in the previous version of the algorithm.

### 4.3. Analysis of the Pattern of the scattered Signal

After determining the centers of the acquired images (Figure 14), we studied the radial profile of the backscattered signal. It showed luminous and dark rings, with a central bright disc concentrating about 60% of the luminous power (Figure 15).

We then applied the developed two algorithms to several images, computed the centers of the patterns, and inferred the radial distances and angular separation (as seen from the sphere) of the dark rings (Figure 16). We observed that:
At each distance, a slow envelope function modulates a higher frequency oscillation (Figure 15). This latter turns out not fully periodical, as the ring mutual distances decrease outwards.The pattern increases in size with increasing distances, revealing a divergence of the backscattered beam, at least in the far field.This divergence does not exhibit the same angle (for a specific ring) at all distances. Figure 16b shows an increase of the dark ring angular separation with the distance. If each dark ring is regarded as the base of a cone with apex in the target sphere, then the apex angle increases with the sphere distance, as in a sort of angular amplifier. It would remain constant instead if the backscattered beam were shaped as a series of coaxial cones. This is in contrast with one of the most accepted models for the glory found in literature [17]. This theory asserts that the backscattered wavefront has a toroidal shape whose intensity is a function (linear combination of squared Bessel functions of first kind) of a parameter $u=\frac{2\pi an}{\lambda }\gamma$, where *a* is the radius of the sphere, *n* the sphere refractive index and γ. is supplementary to the backscattering angle (Figure 17). There is no dependance on the distance, so the divergency angle *u* is expected to be the same at all distances with no near/far-field separation.

### 4.4. Comparison with “Classical” Software and Performance Analysis

We compared the developed algorithms with estabilished software (Classes *cv2.moments* and *scipy.ndimage.center_of_mass* and *ImageJ* with its measurement tool.). The derived image center after segmentation was the base for comparison. We carried out some tests:
(1)*Center repeatability vs. distance of the camera to the sphere*. This test is based on simulation only. We used Zemax [20] as simulation software. Figure 18 shows the used layouts, Figure 19 some examples of the simulated backscattered images, Figure 20 the size of a surrounding ring as a function of the distance to the sphere. Figure 20 shows a minimum, which separates the converging (near-field) from the diverging (far-field) portions of the beam, thus being a “waist”. The derived centers were compared with the theoretical one imposed by simulation. The discrepancy $\Delta \;=\sqrt{\delta x^{2}+\delta y^{2}}$ is a measure of the error incurred by each algorithm at each distance (Figure 21). The simulation allowed to move the target over an ideal straight line parallel to the camera optical axis, theoretically resulting in the same image coordinates of the centers. This would have been impossible in practice—or very difficult and expensive—over distances of several meters. The pooled standard deviations of the derived center *x* and *y* coordinates over the full distance range (100–5000) mm were: 0.52 and 0.51 pixel for variance-based and FFT-based algorithms respectively; 0.69 pixel for the *scipy.ndimage.center_of_mass* algorithm; and an impressive 0.03 pixel for the ImageJ software. In ideal conditions (full pattern captured with no border cropping and no noise), ImageJ outstands.(2)*Center accuracy vs. lateral displacement of the sphere.* We then investigated the effect of eccentriciy of the image to the camera and consequent border cropping of the signal; in other words, when the sensor is not used merely for zero detection but for a full measurement of the lateral position. This might be the case e.g., when alignment or tracking is being sought. A same acquired (not simulated) image was shifted in steps to obtain other images corresponding to as many shifts of the sphere. In this case, the developed algorithms are the best choice for achieving the information needed: they are reliable and robust, since they can correctly track the center also when the symmetry of the signal is seriously impaired or totally broken (e.g.,: when an important part of the peripheral rings is cut off the frame); in these situations, the coordinates of the center determined by ImageJ are manifestly wrong. To show this concept, a set of 11 images were cropped along a specific direction (*x*-axis), with each new image separated from the previous one by an arbitrary number of pixel; the error in the determination of the *x*-coordinate of the center has then been calculated and plotted for comparison. In the total absence of a thresholding strategy, when applying ImageJ measurement tool to the cropped (rough) image sequence, the accuracy of VAR and FFT-based algorithms outperforms ImageJ capabilities (Figure 22a). When the correct ImageJ thresholding procedure selects the portion of the signal that keeps its best symmetry and integrity during the lateral displacement of the sphere, the performances of the three methods become comparable (Figure 22b); here, our algorithms still work on the original, not pre-processed images); anyway, also in this case, the variance of the plotted parameter obtained by means of ImageJ is about one order of magnitude greater than the variance of the same parameter obtained using our algorithms: this means that the latter generate coordinates of the centers that are stable and reliable also in presence of images showing patterns with a very low level of simmetry. The artifice that can be used to improve the ImageJ results is, therefore, thresholding the signal with the most appropriate method (that can be different from time to time, according to the pattern, the noise, the number, the thickness and the spacing of the rings…) to isolate its most symmetric portion inside the frame; however, the price, in terms of the amount of the power that is lost in this way, can be high, and the information acquired about the center inaccurate. Without such pre-processing of the eccentric images, on the other hand, ImageJ is very likely to generate macroscopic inconsistencies in the determination of the needed information (Figure 23).(3)*Repeatability of center determination vs. noise of the image*. In this case, the experimentally acquired images were artificially altered by means of Gaussian noise with a gradually increasing variance: a set of 10 images of the sphere at a fixed distance from the sphere, with decreasing *peak signal to noise ratio* was used in a test-like environment, starting from the image with the lowest level of noise (very close to the one effectively acquired by the CMOS camera) to the most noisy one. Figure 24 shows the performances of the developed algorithms in a short sphere-camera distance situation.The developed algorithms do not perform remarkably better than the existing ones for all the investigated distances: orange and grey bars—representative of the *σ*-pooled of the centers found by using VAR and FFT-based algorithms—are, apart from a couple of occurrences, higher than yellow and blue ones—representative of the competitor algorithms—*σ* = 40 is the maximum value of noise tolerated by them; values of *σ* > 40 deteriorate too much the images and generate inconsistencies in the determination of the coordinates of the center. Anyway, it is useful to point out that, in some situations, the centroid algorithm completely failed the test (this is the reason why it is not represented in the plot), while both VAR and FFT algorithms always succeeded in the task of determining the center, even in the cases of heavily degraded images.

Another way of estimating the performance of the algorithms is to test the repeatability of center determination with respect to the choice of the ROI size: the algorithms turn out to be as much robust as they are more capable of generating “similar” center coordinates Cx_i_ and Cy_i_, independently on the ROI size. To verify this aspect (directly correlated to the achievable uncertainty), the quantity $\Delta_{r}=r_{i}-\bar{r}$, where $r_{i}=\sqrt{Cx_{i}^{2}+Cy_{i}^{2}}$ and $\bar{r}=\sqrt{\frac{{!}_{i}\left(Cx_{i}^{2}+Cy_{i}^{2}\right)}{N}}$ has been plotted for a randomly chosen test image (Figure 25).

The distribution of the results obtained seem to limit the best achievable uncertainty to a range of ~±0.5 pixel for the variance-based algorithm and ±0.3 pixel for the FFT-based algorithm. It is remarkable the fact that a small ROI size (compared to the diameter of the central disc, equal to 85 pixel in the image studied) seems adequate to ensure a small value of uncertainty: that allows a fast computation time to determine the center coordinates, even in the case of high-resolution images.

### 4.5. Some Words on Longitudinal Metrology

As stated before, the pattern of the backscattered signal captured by the camera shows a strong dependence on the distance of the camera from the sphere: the ring spacing thickens when the camera is close to the sphere, while it thins out when the camera is far from the sphere. In first approximation, one could be tempted by the possibility of exploiting this behavior to infer some kind of longitudinal metrology on the effective distance of the sphere: let’s see with which level of accuracy. Adding to the algorithms the capability of averaging—along columns—the pixel intensity of the polar-transformed image of the sphere, and then performing the FFT on the row vector of these values, the coordinates of the peaks reveal the average spacing of the bright rings of the original image. The subsequent step is fitting the positions of these peaks with respect to the distance of the sphere, as shown in Figure 26.

The most accurate available points (in a limited range of distances) have been used to fit the data, but, also in this way, the slightly increasing slope of the fitting line prevents from getting a very useful information: solving the fitting equation with respect to the variable *d_sphere-camera_*, the equation:$$d_{sphere-camera}=k\frac{bands\;spacing-A}{B}$$ is obtained, where *A* = intercept and *B* = slope of the line, the distance cannot be resolved at a high degree (e.g.,: submillimetric) of accuracy; indeed, the sensibility *1/B* of the whole system is approximately equal to 6.25 mm/px, too low to confer a metrological valence to this approach of image analysis. Better results could be achieved in presence of a camera with a higher resolution and a bigger active area of the sensor, in order to collect a useful part of the diverging signal at greater distances.

## 5. Conclusions

With this work, the possibility of extracting accurate metrological information about the position of glass spheres, used as targets, with refractive index almost equal to 2 was investigated. More specifically, two completely new algorithms were written to generate the coordinates of the centers of the same spheres through the analysis of their backscattered signals: the complexity of these signals, combined to the lack of a satisfying theoretical approach to interpret those such peculiar patterns observed, forced us to limit our research in extracting quantitative data from empirical observations rather than applying ready-made models. The main strength that these algorithms demonstrated in the validation tests is their ability of working on images with a poor level of symmetry of the pattern caught on the active area of the sensor, a condition that experimentally is quite common, and inferring the desired information without a specific pre-processing (tipically starting from an arbitrary and dangerous thresholding) of the images themselves. Moreover, they proved to be robust and repeatable, also in conditions of heavy noise and degradated images: their accuracy is comparable to the classical software used during the validation tests.

Future developments of this work will concern partly the improvement of the codes efficiency—hopefully taking advantage of the parallel computation through optimized toolkits (like CUDA [21]), very useful when dealing with high resolution images—and on their integration into a real-time acquisition system, partly the efforts to build a suitable mathematical model in order to explain, in a rigorous way, the behavior of the specific retroreflectors we used for target acquisition.

## Figures and Tables

**Figure 1 sensors-19-01082-f001:**
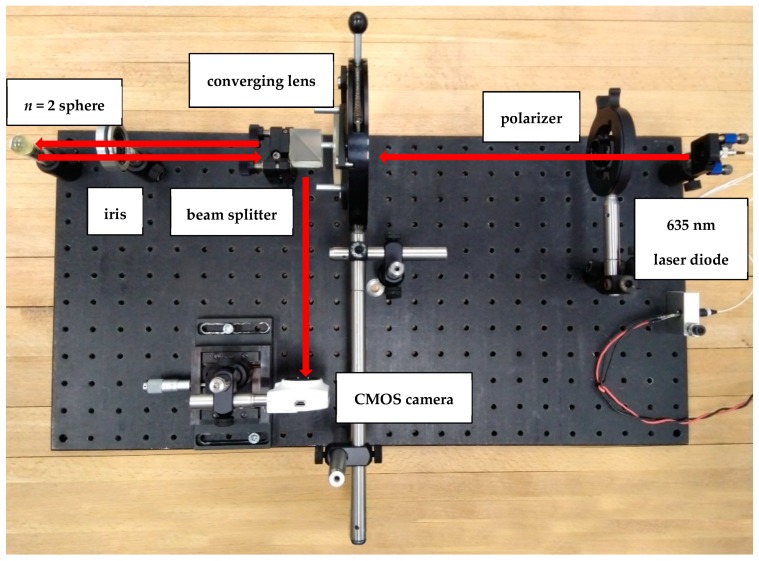
Setup used to study the high-index sphere as retroreflector.

**Figure 2 sensors-19-01082-f002:**
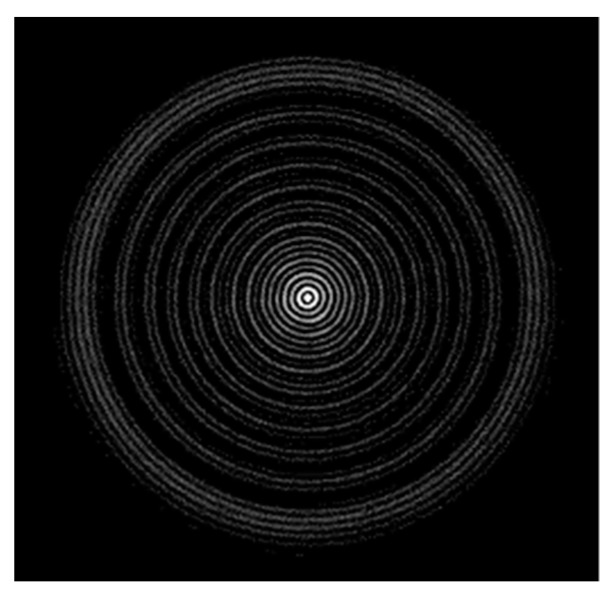
Typical pattern of the backscattered optical signal generated by a S-LAH79 glass sphere.

**Figure 3 sensors-19-01082-f003:**
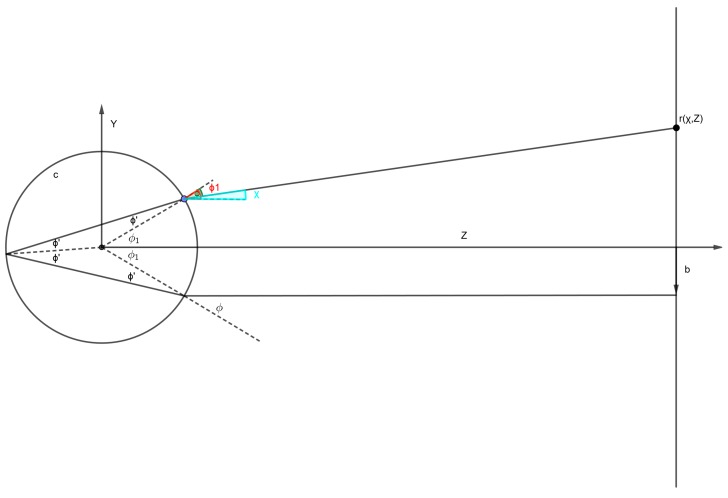
Scheme with the most important geometrical parameters for simple ray tracing.

**Figure 4 sensors-19-01082-f004:**
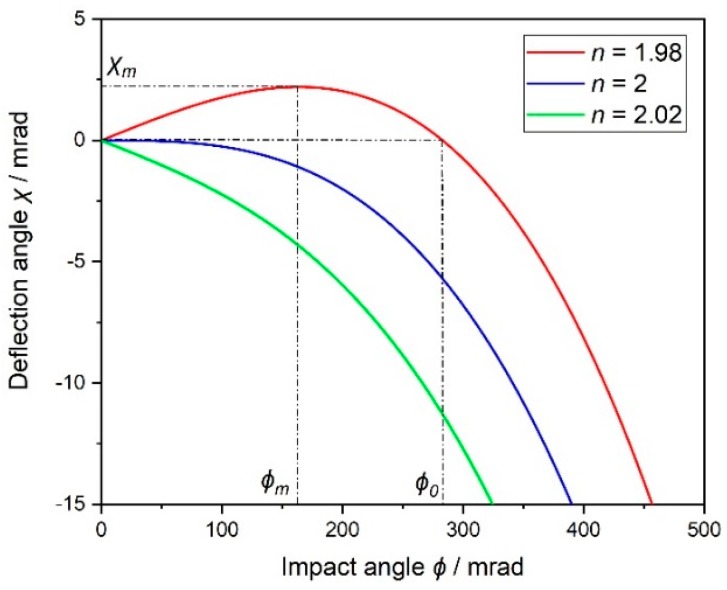
Transfer function for incident and reflected rays of different refractive indices n (the red curve is the most representative for our situation); as soon as a ray moves away from Z axis, increasing its impact parameter *b*, the deflection angle increases in this way: from a null value, it reaches a maximum in correspondence of $\phi_{m}$, then it starts decreasing until a null value (parallel to itself but mapped in the other half-space) when $\phi =\phi_{0}$. Image inspired by [5] (Figure 3).

**Figure 5 sensors-19-01082-f005:**
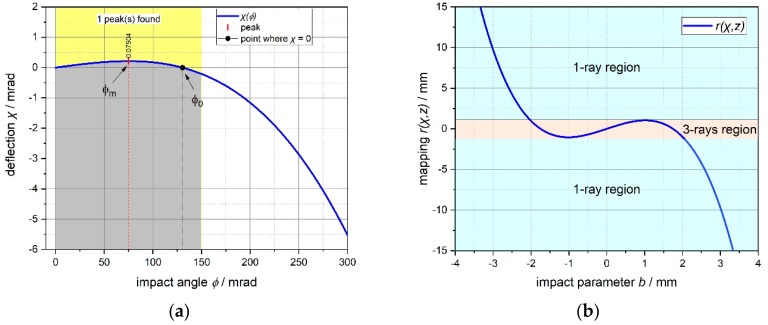
(**a**) Deflection angle *χ* for the glass used, depending on the impact angle *ϕ*. (**b**) Mapping ***r*** of the ray exiting the sphere as function of the impact parameter ***b*** (when z = 1 m); the plot is windowed where |*b*| < *R*/2 (here the most general expression r(χ,z)=Rsin(ϕ+χ)+[z−Rcos(ϕ+χ)]tanχ was used).

**Figure 6 sensors-19-01082-f006:**
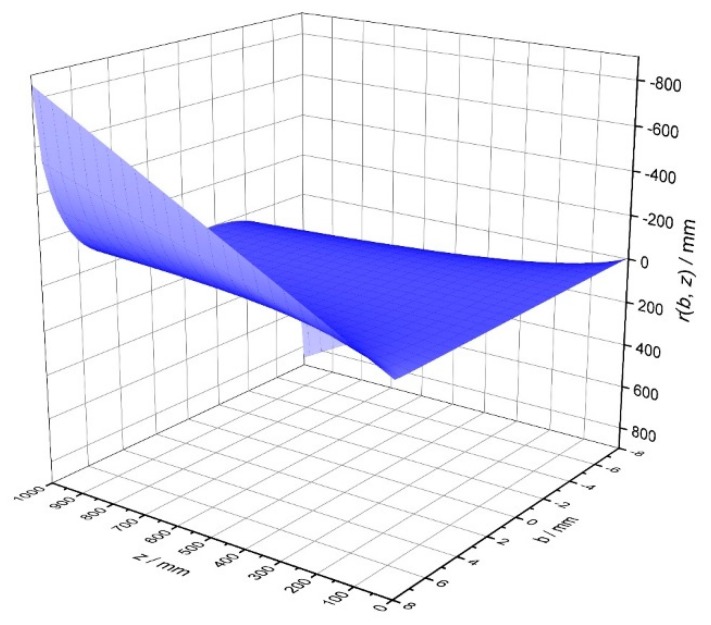
3D plot of the sphere transfer function with respect to the impact parameter **b** and the distance **z** from the screen.

**Figure 7 sensors-19-01082-f007:**
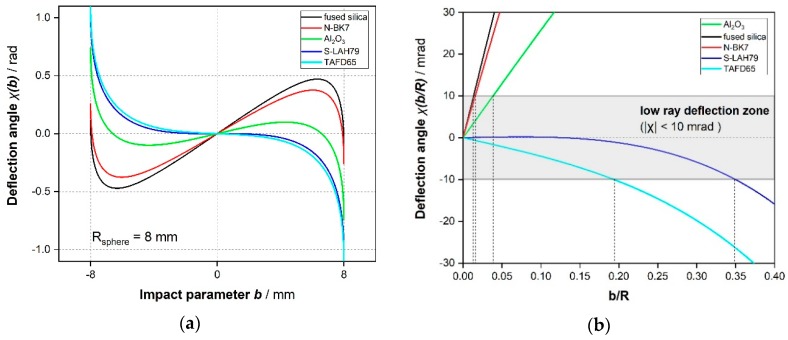
(**a**) Deflection angles plot for five types of glass, depending on their impact parameter (*R* = 8 mm is fixed for all the spheres). (**b**) Highlight on low ray deflection zone for the same glasses, depending on the normalized impact parameter.

**Figure 8 sensors-19-01082-f008:**
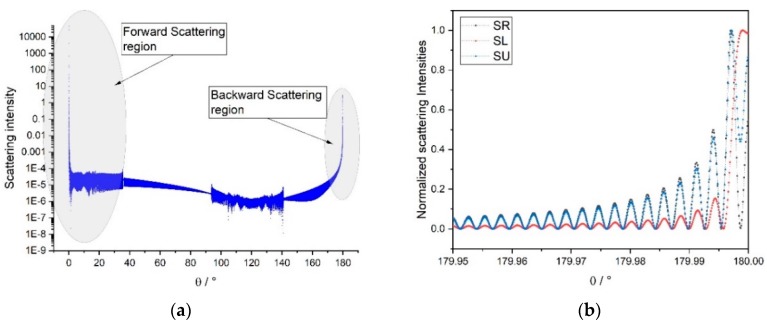
(PyMieScatt package). (**a**) Scattering intensity pattern in the whole (0–180)° angular domain for a 16 mm S-LAH79 glass sphere, with *λ* = 635 nm. (**b**) Focus on the backward scattering region for the same sphere.

**Figure 9 sensors-19-01082-f009:**
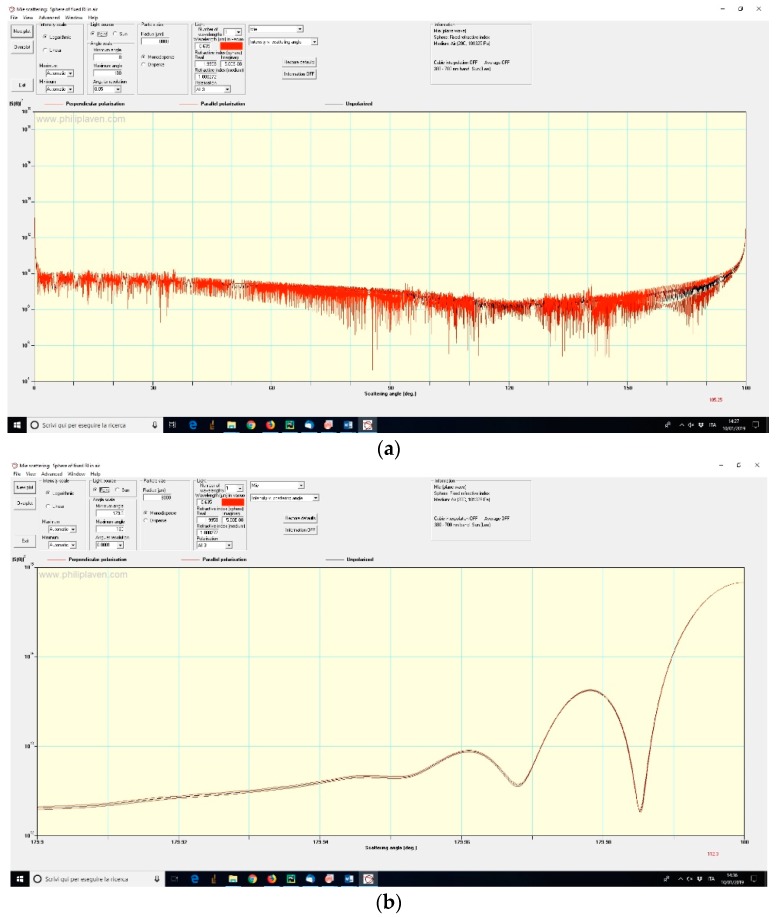
(MiePlot software). Replica of simulation conditions (same as Figure 8); in (**a**) the whole angular range is represented, while in (**b**) there is an emphasis on near-180° region (in logarithmic scale).

**Figure 10 sensors-19-01082-f010:**
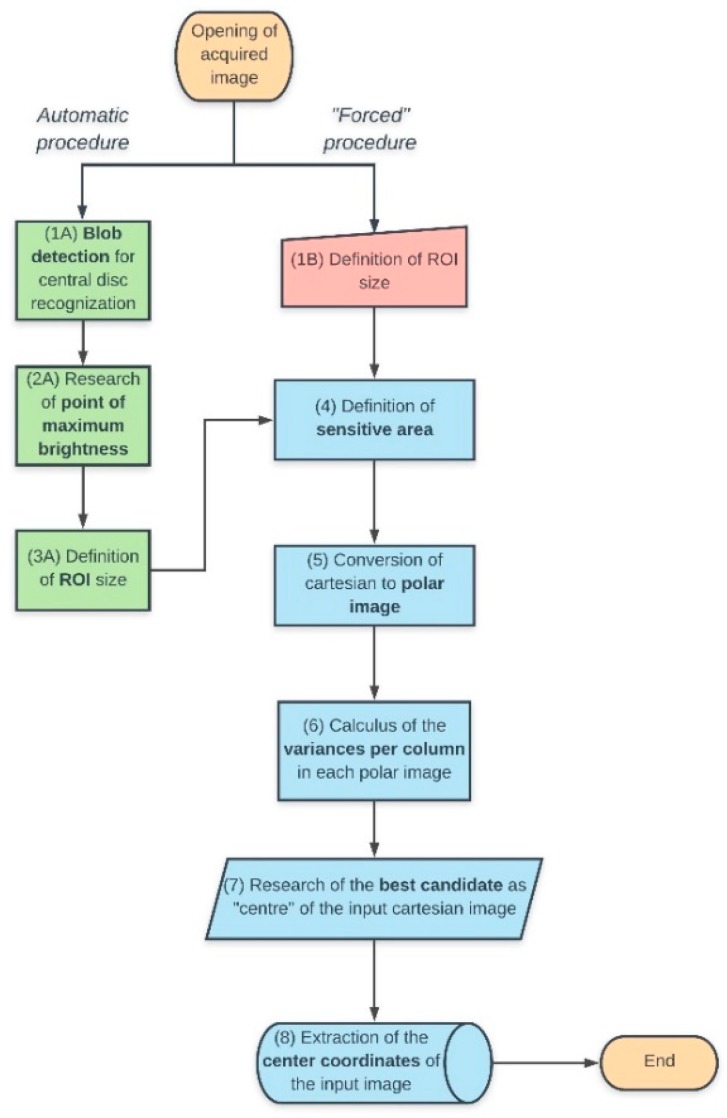
Block diagram of the variance-based approach to the problem.

**Figure 11 sensors-19-01082-f011:**
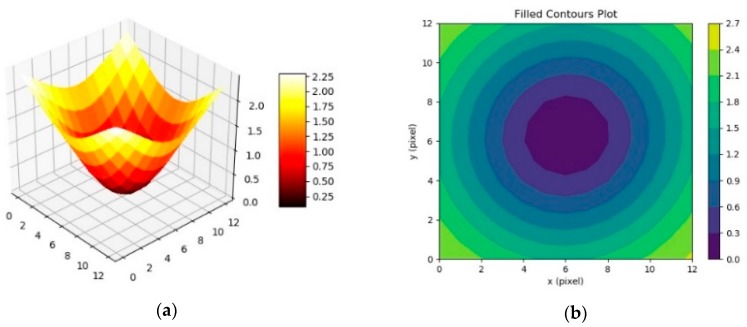
(**a**) 3D surface obtained by plotting the matrix of the sum of the variances. (**b**) Contours plot of the 2D projection of the same matrix.

**Figure 12 sensors-19-01082-f012:**
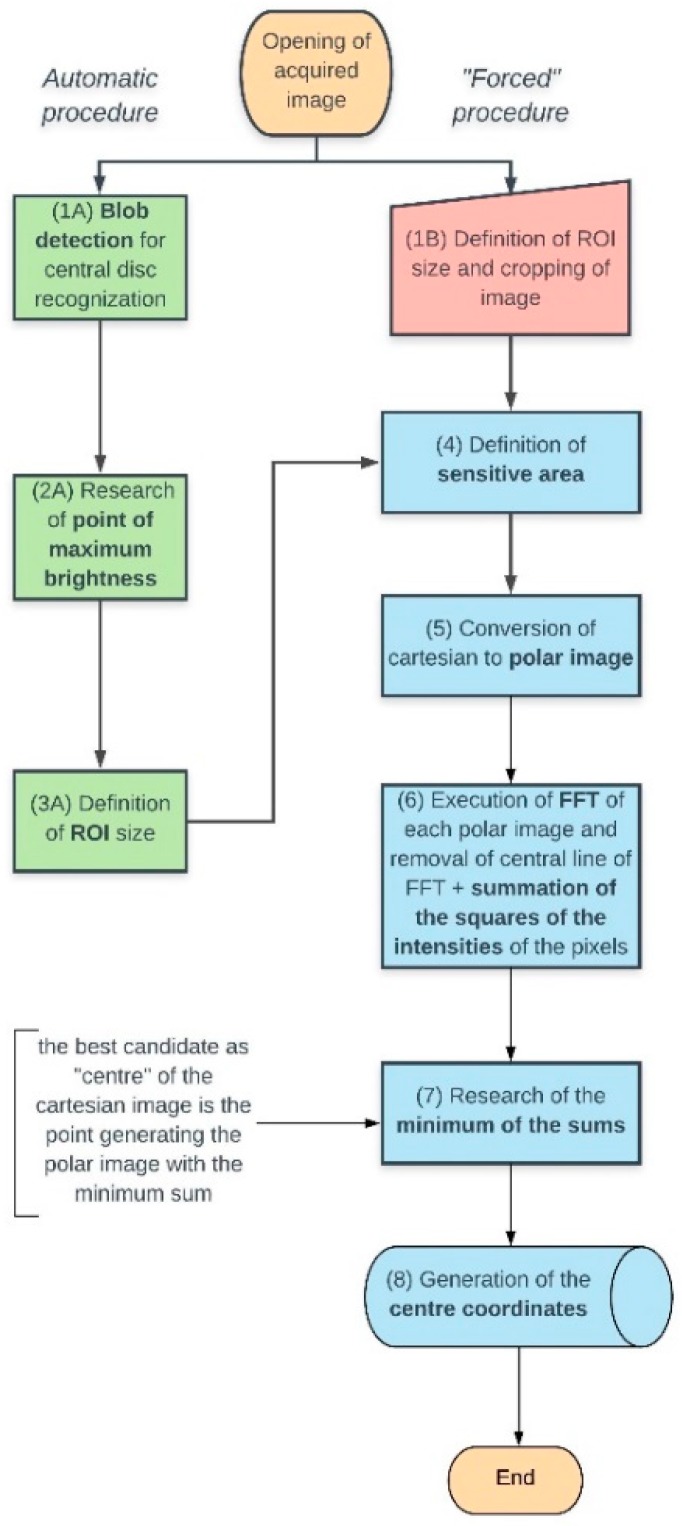
Block diagram of the FFT-based approach to the problem.

**Figure 13 sensors-19-01082-f013:**
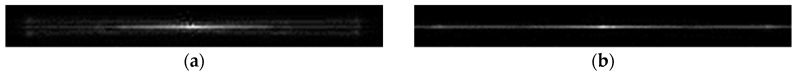
(**a**) Example of power spectrum obtained by performing FFT on one of polar-transformed images. (**b**) Best power spectrum of the polar-transformed images, obtained in correspondence with the pole which gives the best symmetry.

**Figure 14 sensors-19-01082-f014:**
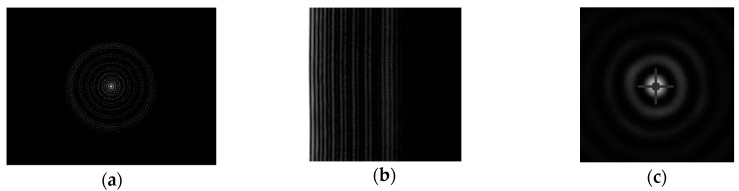
Working sequence of the algorithms: input image (**a**); best polar plot (**b**); center with minimum sum of variances (VAR-basef algorithm) or minimum sum of pixel intensities (FFT-based algorithm) (**c**).

**Figure 15 sensors-19-01082-f015:**
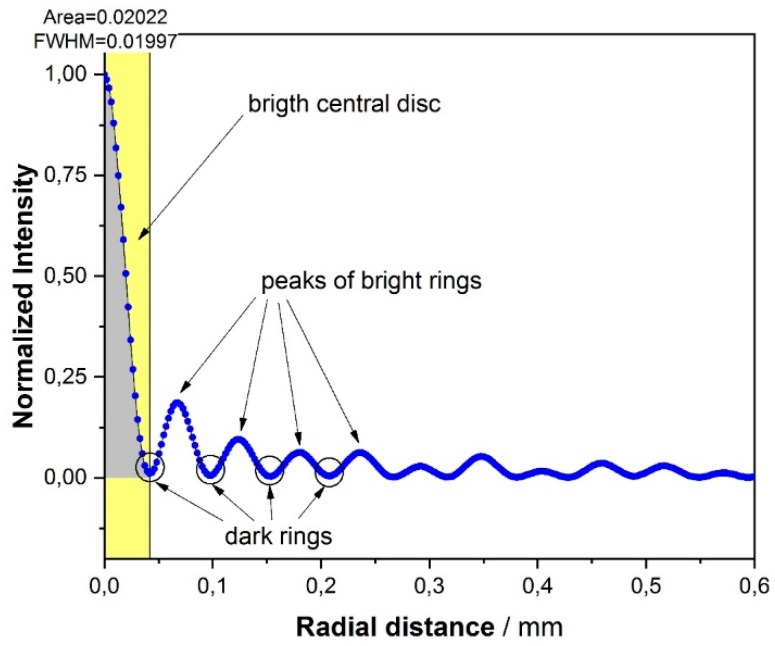
A typical intensity pattern of the backscattered image, starting from the center found by using one of the algorithms and moving towards the periphery of the same image.

**Figure 16 sensors-19-01082-f016:**
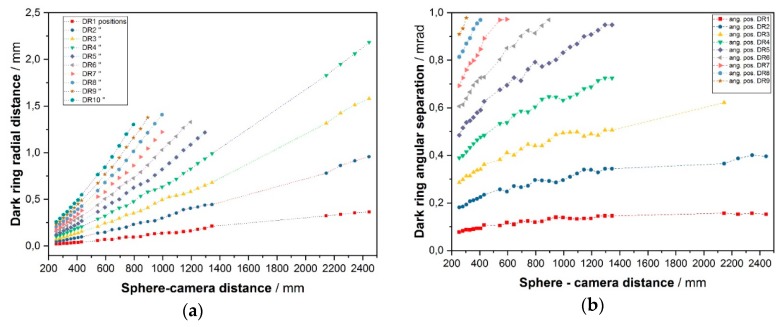
Dark ring radial distances (**a**) and corresponding angular separation as seen from the sphere (**b**), as a function of the distance to the sphere. Dark rings are numbered outwards: DR1, …, DR*n*.

**Figure 17 sensors-19-01082-f017:**
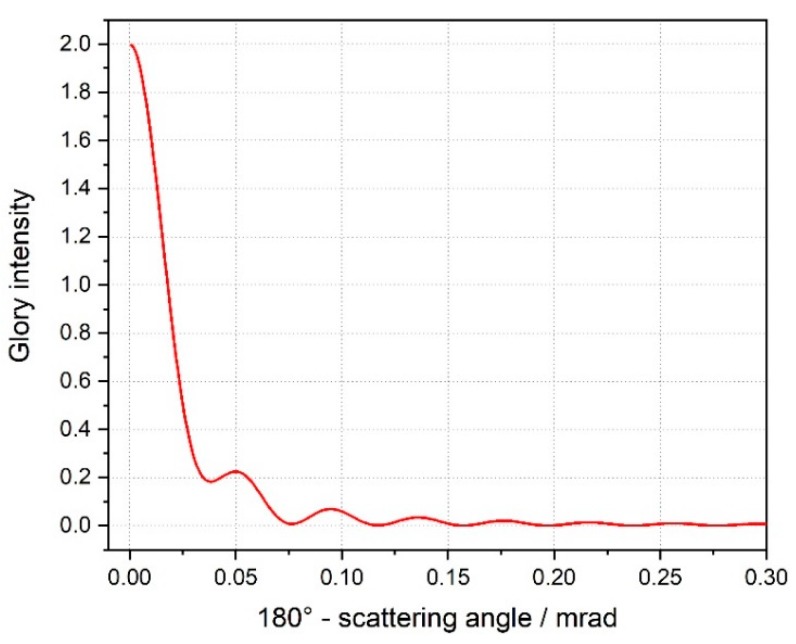
Intensity of the glory as predicted in [17] (with C1 = 0, C2 = 1).

**Figure 18 sensors-19-01082-f018:**
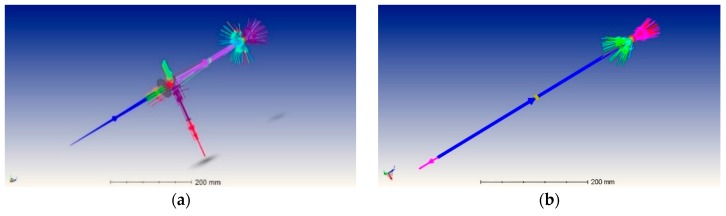
Two equivalent set ups simulated with Zemax. (**a**) Complete layout with all optical components; (**b**) Functionally equivalent simplified layout (to improve the simulation speed).

**Figure 19 sensors-19-01082-f019:**
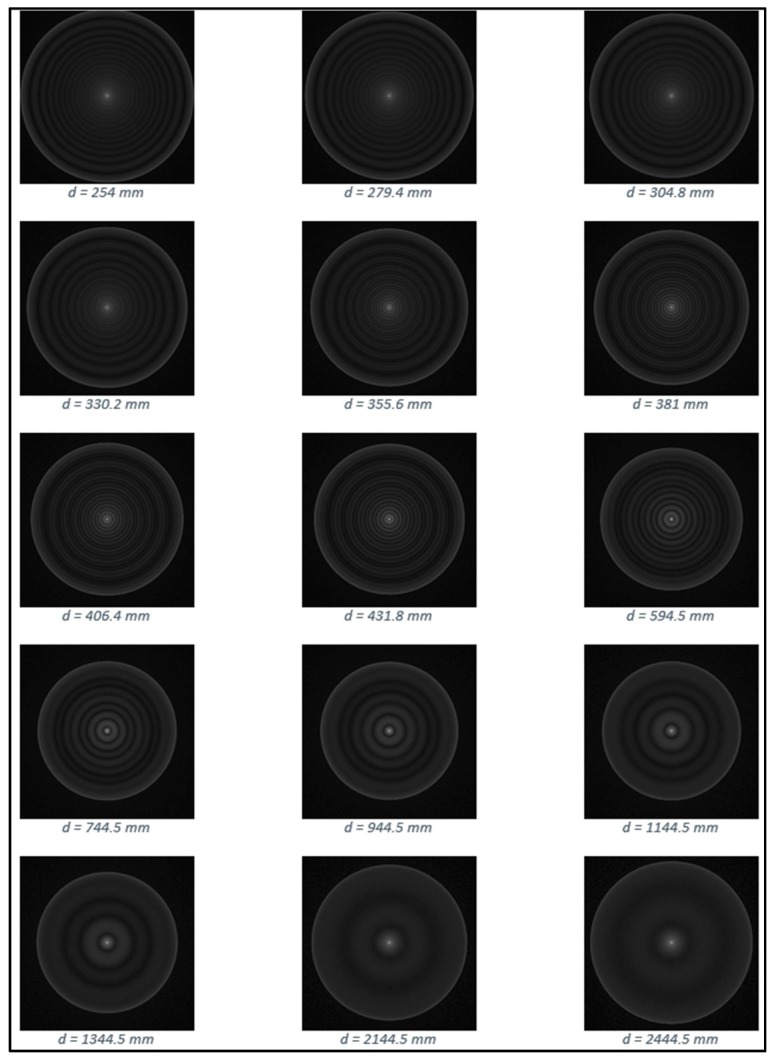
Some backscattered simulated images at different distances to the sphere. Contrast enhancement is used to enhance the signal.

**Figure 20 sensors-19-01082-f020:**
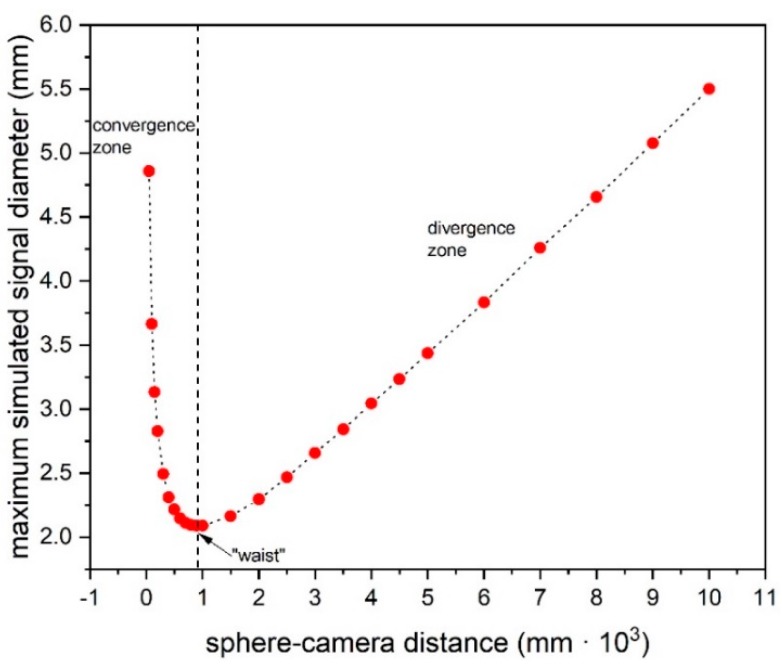
Size of the surrounding ring at different distances to the sphere.

**Figure 21 sensors-19-01082-f021:**
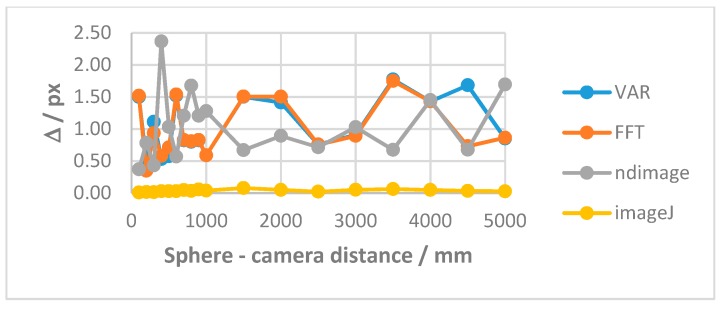
**Δ** vs. sphere-camera distance.

**Figure 22 sensors-19-01082-f022:**
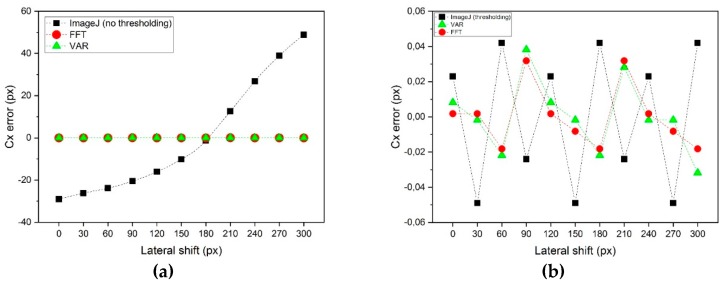
Error in the determination of the x-coordinate of the center as a function of the lateral shift of the sphere, when the sphere-camera distance is ~ 550 mm. (**a**): without ImageJ thresholding; (**b**): with ImageJ thresholding.

**Figure 23 sensors-19-01082-f023:**
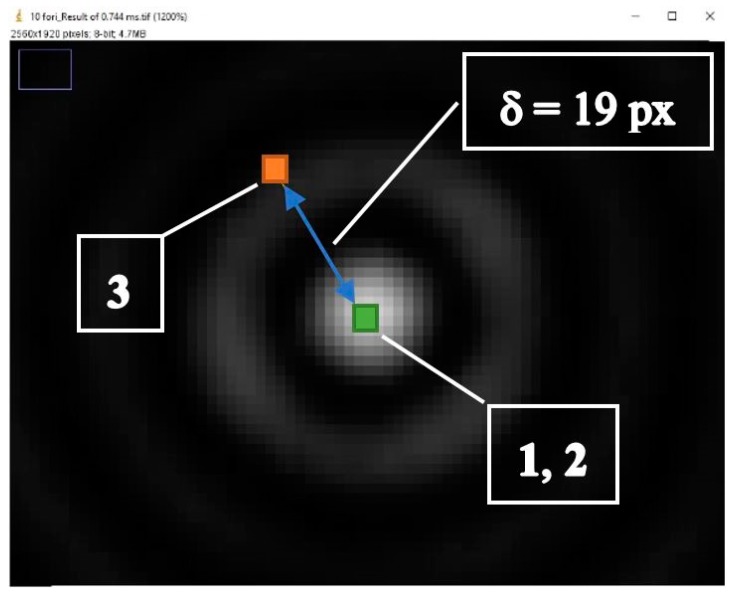
Example of ImageJ center determination “failure” without a proper thresholding of the image: nearly overlying labels ‘**1**’ and ‘**2**’, inside the central bright disc, show the centers found by VAR and FFT-based algorithms, while label ‘**3**’ shows the ImageJ result; the distance δ between the centers is ~42 μm in this test configuration! (here the pixel size is 2.2 μm).

**Figure 24 sensors-19-01082-f024:**
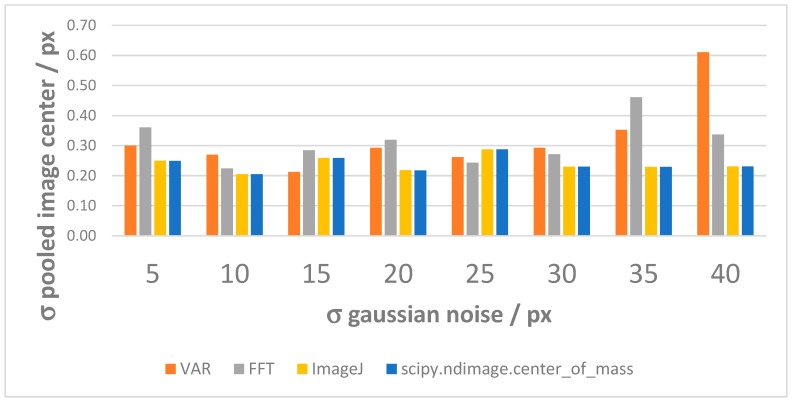
Repeatability of center determination vs. image noise, tested with different algorithms: a representative example (here the distance of 254 mm is investigated).

**Figure 25 sensors-19-01082-f025:**
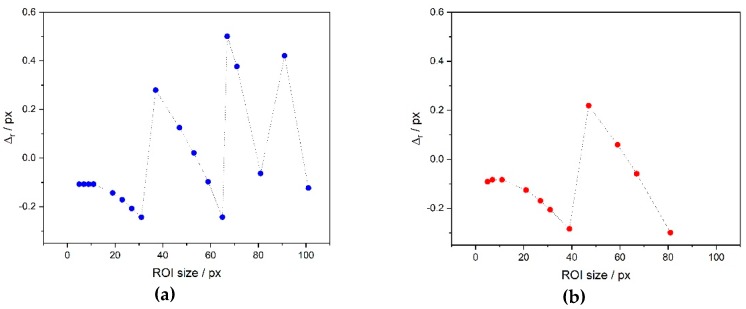
$\Delta_{r}$ for variance (**a**) and FFT-based (**b**) algorithms.

**Figure 26 sensors-19-01082-f026:**
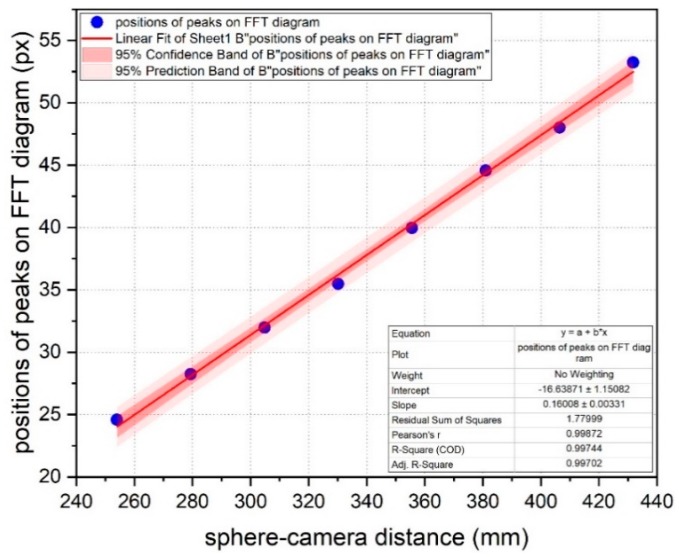
Average spacing (in pixel) of the bright bands of the polar back scattered image of the sphere; the spacing of the bright rings of the cartesian image is connected to the vertical bands spacing through a factor *k* dependent on the resolution of the original image.

**Table 1 sensors-19-01082-t001:** Fundamental equations for tracing rays of light impacting on a sphere (taken from [5]).

Relation between ray (coming from the right side of the sphere) impact parameter b and impact angle *ϕ* (with *R* = radius of the sphere):	b=Rsinϕ
Deflection angle of the ray exiting the sphere (Figure 4):	$\chi =4\sin^{-1}\left(\frac{\sin\phi }{n}\right)-2\phi$
Position *r* of the point (on a hypothetical screen at distance *z* from the sphere bigger than its radius) mapping the input ray:	r(χ,z)≅Rsin(ϕ+χ)+zsinχ.
Maximum deflection angle corresponding to the impact angle *φ**_m_* (when *z* ≫*R*):	$\chi_{m}=4{\left(\frac{2-n}{3}\right)}^{\frac{3}{2}}\;\mathrm{for}\;\phi_{m}=2\sqrt{\frac{2-n}{3}}\;$
Condition of perfect (when *χ* = 0) back-reflection:	$\phi =\phi_{0}=2\sqrt{2-n}=\sqrt{3}\phi_{m}\;$
Beam intensity distribution on a screen at distance *z* from the sphere:	$(r,z)=f[b(r)]\frac{bdb}{rdr}\approx$ $f\left[Rsin\phi \right]\frac{R^{2}\sin2\phi }{z\frac{d}{d\phi }{\left[R\sin\left(\phi +\chi \right)+z\sin\chi \right]}^{2}}\;$
Needed aperture radius of the impinging beam to have the maximum acceptance of the sphere:	$R_{acc}=R\phi_{-\chi_{m}}=\frac{2R}{\sqrt{3}}\phi_{0}$

**Table 2 sensors-19-01082-t002:** list of main optical parameters considered in ray tracing simulations.

Input Parameters:	*z*/mm	*n*	*R*/mm	*b*/mm		
	1000	1.9958	8	0.5		
Parameters depending on *R*, *b*, *n*:	ϕ	*χ*/mrad	*r(χ, Z)*/mm			
	62.54	0.20	0.70			
Parameters depending only on *R* and *n*:	*ϕ****_m_***/mrad	***χ_m_***^1^/mrad	***b_m_***/mm	*ϕ****_0_***/mrad	***b_0_***/mm	***R_acc_***/mm
	75.06	0.21	0.60	130.06	1.04	1.20
First three *rainbow* ^2^ angles:	***n_r_***	*ϕ****_rainbow_***/rad	***χ _rainbow_***/rad			
	1	0.08	3.14			
	2	0.91	0.62			
	3	1.11	1.64			

^1^: The expression used here for χm is the more general.
χm=±4sin−1(1n4−n23)∓2sin−1(4−n23) ^2^: Scattering angles at which the intensity is infinite, according to geometrical optics [9]; impact angles ϕ determininig rainbows, after undergoing *n_r_* reflections inside the sphere:cosϕ=n2−1nr(nr+2)

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
