# Peer review of "High-Index Glass Ball Retroreflectors for Measuring Lateral Positions"

_sensors, 2019, doi:10.3390/s19051082_

Round 1
Reviewer 1 Report
If authors can present some points more clearly, the paper is worthy of being recommended as accepted
1. Why does the lateral position of sensors in space needs be measured?
2. Does the simulation software used in the study be competent for coherent
light ray tracing?
3. Does the diffraction effects incurred by the small ball spheres
should not be considered?
4. Can some real optics experimental pictures be shown in the paper for approving
that the idea of measuring lateral position of sensors by glass spheres can
work ?
Author Response
Response to Reviewer 1 Comments
Point 1: Why does the lateral position of sensors in space needs be measured?
Response 1: As explained in the Introduction, the need of lateral position measurement came from the need of improving the performance of a novel coordinate measurement concept (InPlanT [8]). In it, each coordinate is measured by a separate device implementing a coordinate axis. Each device detects the target without contact by means of a camera serving as zero detector, while the actual axis coordinate is measured by a separate linear encoder. Sub-pixel accuracy of the sphere center coordinates was essential for satisfying the target uncertainty. Further to the original intent, the non-contacting detection/measurement of later positions is possibly of wider use.
We added a paragraph in the Introduction to hopefully explain this better.
Point 2: Does the simulation software used in the study be competent for coherent light ray tracing?
Response 2: The ray tracing software used, Zemax OpticStudio, is perfectly suitable for such analysis: it is one of the most representative programs commercially available today for optical design; it proved to be reliable and versatile, and particularly is for coherent light, which is a standard problem in Optics (for example, here and here, and also in the dedicated forum).
Point 3: Does the diffraction effects incurred by the small ball spheres should not be considered?
Response 3: For spheres much larger than the wavelength (16 mm in our case), diffraction effects impact mostly on forward scattering phenomena (i.e. θ ≈ 0, direction of the light propagation, see [17] chap. 8 and 13.3). Backscattering - we are interested in - occurs when θ ≈ 180°, and in this case the dominating phenomena are reflection and refraction. We evaluated the diffraction effects, but we decided they should be deemed negligible in our situation.
Point 4: Can some real optics experimental pictures be shown in the paper for approving that the idea of measuring lateral position of sensors by glass spheres can work?
Response 4: We are not sure to interpret you question properly.
If the question is about the experimental set up used for our investigation, this is depicted in Figure 1 of the paper. The set up proved effective and worked in delivering the experimental results.
If the question is instead about the possible use of lateral detection/measurement in general, then the answer is that this can be used in several applications. We originally developed a concept whose working principle is described in [8] and a set up is reproduced in the pictures linked below. It was successfully implemented and tested in harsh conditions, during a measurement campaign. The details of this set up is being described and will hopefully be published soon. See also our answer to your question #1.
picture 1 (please see the attachment)
picture 2 (please see the attachment)

Reviewer 2 Report
The article deals with one partial problem in the metrology field of determining the position of the center of the glass ball. The issue of light reflection from the dielectric sphere has been known for more than 150 years, and many articles and many books have been published about it. The authors of the article describe completely new algorithms for determining the coordinates of the sphere center through the analysis of their backscattered signals.
The article can be considered as another contribution to the issue of large scale metrology and therefore I recommend publish the artikle as it.
Author Response
No response needed.
Reviewer 3 Report
The authors developed a high-index glass ball retroreflector for the lateral position measurement. The explanation of this manuscript is not enough concise and clear. Please see below for specific comments.
1. The purpose of this study is the measurement of the lateral position. The experiments shown in figure 16, 20 and 26 demonstrate the relationship of the sphere-camera distance and some kind of parameters. In my opinion, the sphere-camera distance is a longitudinal distance, not a lateral one. There is no theory and experiment about the lateral position in this manuscript.
2. Authors used the ray-tracing approach and Mie scattering theory to predict the pattern of the backscattered optical signal generated by the glass sphere but in vain. I don’t think it is suitable to describe the useless theories in the manuscript unless authors emphasize their values.
3. What is the diameter of S-LAH79 glass sphere? According to the setup in figure 1, I estimate its diameter at about 1 cm. The center-meter scale is far away from the Mie scattering range.
4. There are python source codes on page 9, 10 and 12. I don’t think it is suitable to describe the source codes in the manuscript.
5. I suggest the author consider the effect of the diffraction.
Author Response
Response to Reviewer 3 Comments
Point 1: The purpose of this study is the measurement of the lateral position. The experiments shown in figure 16, 20 and 26 demonstrate the relationship of the sphere-camera distance and some kind of parameters. In my opinion, the sphere-camera distance is a longitudinal distance, not a lateral one. There is no theory and experiment about the lateral position in this manuscript.
Response 1: The proposed sensor is not intended for measuring the distance of the sphere, and is limited to its lateral position. However, the distance is a very important parameter: the measurement with a nearby sphere is much simpler than for a remote one. This is because the signal intensity decreases with the distance (due to the divergence) and its size increases up to the point of exceeding that of the camera active area, resulting in an unwanted crop. For this reason, all tests were carried out at different distances, to evaluate the performance over a target range. Only in the test whose results are shown in Figure 22 (replaced to better show the concept, in a paragraph that has been improved), the distance does not vary. This test is aimed at evaluating the performance when the target is slightly shifted aside at the same distance. We chose a distance of 550 mm and did not repeat the test at others.
Point 2: Authors used the ray-tracing approach and Mie scattering theory to predict the pattern of the backscattered optical signal generated by the glass sphere but in vain. I don’t think it is suitable to describe the useless theories in the manuscript unless authors emphasize their values.
Response 2: We spent a significant effort in validating the existing theories in the attempt to deeply understand and master the complex observed patterns. We found no previous published similar attempts – which would have helped us. The deviations between the predicted results and the observations are reported here as a possible help for whoever would like to commence a similar exercise in future. Further, these deviations forced us to another approach merely based on the intrinsic symmetry of the image. Describing our investigations also introduces this work providing the background.
Point 3: What is the diameter of S-LAH79 glass sphere? According to the setup in figure 1, I estimate its diameter at about 1 cm. The center-meter scale is far away from the Mie scattering range.
Response 3: The diameter of the used S-LAH79 sphere was 16 mm. This value was reported in Figure 7 (a) and in the caption of Figure 8; we made it more explicit now right at the beginning of chapter 2.
The Mie theory is not limited by the particle size. The entire 12th chapter in [17] is dedicated to the case of very large spheres. A remarkable outcome is the demonstration that it is possible ‘To derive from the Mie solutions a set of formulae identical to those following from ray optics.’ (chapter 12.35). The Mie equations cannot be solved but numerically, whose computational effort is related to the particle size. Even small or medium particles were used to require unaffordable computing time up to few years ago. The sub-millimeter range was used to be a de facto limit to this approach. Nowadays’ computers have enough power to solve those equations in a couple of hours, also for as big spheres as the ones we used.
Point 4: There are python source codes on page 9, 10 and 12. I don’t think it is suitable to describe the source codes in the manuscript.
Response 4: Less than 50 lines of code are reported, split in the two proposed algorithms. We think that the code itself has an essential role in illustrating the algorithms, in conjunction with the explanatory text and figure. We considered the option of moving it to an appendix, but it turned out that the needed description in the main body would have been either insufficient – and then confusing – or hardly understandable without the code.
Point 5: I suggest the author consider the effect of the diffraction.
Response 5: For spheres much larger than the wavelength (16 mm in our case), diffraction effects impact mostly on forward scattering phenomena (i.e. θ ≈ 0, direction of the light propagation, see [17] chap. 8 and 13.3). Backscattering - we are interested in - occurs when θ ≈ 180°, and in this case the dominating phenomena are reflection and refraction. We evaluated the diffraction effects, but we decided they should be deemed negligible in our situation.

Round 2
Reviewer 3 Report
I agree and accept all of the authors’ responses. However, I still concern response 5. Because the refractive index of the sphere is high, the back-reflections from the two surfaces of the sphere (air-sphere and sphere-air) should be considered. This point is the same as the authors' viewpoint. Does the interference from these two reflections should contribute to the pattern? In summary, this manuscript can be accepted.